# Synergetic Effect of Chemical Coagulation and Electroflotation on Synthetic Palm Oil Mill Effluent Treatment

**Enjeh Yoland Fobang, Takeshi Fujino \* and Thenuwara Arachchige Omila Kasun Meetiyagoda \***

Department of Environmental Science and Technology, Graduate School of Science and Engineering, Saitama University, 255 Shimo-okubo, Sakura-ku, Saitama 338-8570, Japan
\* Correspondence: fujino@mail.saitama-u.ac.jp (T.F.); 822kasun@gmail.com (T.A.O.K.M.)

**Abstract:** Palm oil mill effluent (POME) is considered the most environmentally harmful when discharged without proper treatment. In addition to conventional biological treatment methods, physicochemical treatment techniques are considered alternative methods to treat POME as polishing or post-treatment techniques to meet the discharge water quality standards set by authorities. Recently, electroflotation (EF) has gained popularity in wastewater treatment owing to its high efficiency, no harmful by-products, and ease of operation. However, EF has limitations on energy consumption because high current density and long electrolysis time are often used to increase the density of gas bubbles and metallic ions produced in the EF system used in pollutant removal. Polyaluminum chloride (PAC) and cationic polyacrylamide (CPAM) are used as alternative options for the production of coagulants instead of using a sacrificial anode in EF. In this study, we hypothesized that PAC and CPAM could enhance the efficiency and reduce the specific energy consumption of EF by minimizing the electrolysis time used in POME treatment. The effects of electrolysis time, current density, and coagulant dosage on POME treatment were investigated. EF treatment at a current density of 2.5 mA/cm$^2$ has achieved 82% of turbidity and 47% of chemical oxygen demand (COD) removal after 45 min electrolysis time, consuming 0.014 kWh of specific energy for the treatment of one gram of COD. There was no improvement in terms of turbidity removal when the current density was increased from 2.5 to 5 mA/cm$^2$; however, the COD removal efficiency was increased up to 52% at 5 mA/cm$^2$. When EF was performed at 1 A combined with PAC at a dosage of 40 mg/L and CPAM at a dosage of 20 mg/L, it was noticed that turbidity and COD removal increased up to 96% and 54%, respectively, within 15 min electrolysis. Subsequently, the specific energy consumption was reduced to 0.004 kWh (by 71%) per one gram of COD treatment. Results confirmed that the chemical coagulants could increase the POME treatment efficiency and reduce the specific energy consumption of EF. However, this method can be improved aiming at further reduction of COD by mineralizing the dissolved organic compounds to fulfill the POME discharge quality standards.

**Keywords:** palm oil mill effluent; chemical coagulation; polyaluminum chloride; cationic polyacrylamide; electroflotation; synergetic effect; specific energy consumption

## 1. Introduction

Indiscriminate disposal of high-strength organic pollutants poses a threat to the surrounding water bodies, leading to eutrophication and endangering aquatic life [1]. Palm oil mill effluent (POME), which contains large quantities of organic matter, is considered the most environmentally harmful when discharged untreated [2,3]. Palm oil is a popular edible vegetable oil that is extracted from the fruits of oil palm trees (*Elaeis guineensis*). Oil palm trees are native to West Africa; however, they were introduced to South-East Asian countries such as Malaysia, Indonesia, and Thailand, and currently, those countries contribute to producing over 85% of the global supply [4]. The demand for palm oil is rapidly increasing owing to its positive health impacts, such as increasing brain health, reducing oxidative stress, improving hair and skin health, etc. [5], while some studies

reported that the consumption of palm oil increases low-density lipoprotein cholesterol [6]. The production process of palm oil resulted in the release of extensive amounts of liquid waste known as POME, which is a significant concern [7]. During industrial processing, every ton of crude palm oil can produce approximately 2.5–3.8 tons of POME [7,8].

POME normally consists of more than 95% of water, 4–5% of total solids, including 2–4% of suspended solids, and 0.6–0.7% of residual oil [5,9,10], which has a thick brownish color, viscous appearance, high concentration of colloidal suspension, and acidic properties with a nuisance odor [11]. Most palm oil industries discharge POME into the environment or directly to water sources without proper treatment. Numerous negative environmental and sociological impacts are associated with partially treated POME discharges. POME is composed of complex compounds, which are difficult to break down and associated with a high concentration of chemical oxygen demand (COD). However, some components of POME are water soluble and biodegradable, rapidly increasing the biochemical oxygen demand (BOD) in water. The microbial population increases proportionally to the amount of available biodegradable organic matter in POME and rapidly consumes oxygen, which leads to reduced dissolved oxygen in water. Subsequently, hypoxia or anoxic conditions are developed. In this case, other aquatic organisms would no longer survive in an aquatic environment and disrupt ecosystem balance [2–4]. Furthermore, oil and grease (O&G) in POME may have some potential impacts on aquatic environments [2]. Even though POME is non-poisonous to humans [12], many sociological concerns can arise, for example, owing to improper disposal of solid waste and POME, which can be a nuisance to the residents living close to the palm oil mills [13] and pose a threat to the surrounding water bodies and soil properties [2,3]. Therefore, POME should be appropriately treated up to standards before releasing them into the environment. Table 1 lists the discharge quality standard of POME enforced by the Department of Environment (DOE), Malaysia, under the regulation of crude palm oil, 1982 [14].

**Table 1.** Discharge quality standards of POME (DOE, 1982).

| Parameter | Limit |
|---|---|
| pH | 5.0–9.0 |
| Temperature | 45 °C |
| Total suspended solids | 400 mg/L |
| BOD (3-day) | 100 mg/L |
| COD | N/A |
| Oil and grease | 50 mg/L |
| Ammonia as nitrogen | 150 mg/L |

Iskandar et al. [15] compared the standards for POME discharge in different countries. The maximum levels of COD were 350 and 120 mg/L for Indonesia and Thailand, respectively. In Malaysia, the permissible limit of COD was 1000 mg/L, which was used from July 1981 to May 1982. However, there has been no new discharge limit for COD stipulated since then [14,16].

In POME treatment, pretreatment is the primary step and is performed by mesh screen or vibration to remove coarse particles. Then, the oil is separated by oil skimmers or an aeration process. As the main treatment process of POME, most palm oil manufacturers prevalently use open ponding systems, which consist of anaerobic ponds, facultative ponds, and aerobic lagoons [5,17,18]. For example, 85% of POME treatment in Malaysia is carried out using the conventional ponding system because it has a low operating cost and is more convenient. The biological treatment can easily apply if the biodegradability of POME is higher than 0.5, which is a ratio of BOD and COD [4]. However, lagoon treatment ponds require extensive land area, long retention times (for anaerobic ponds, hydraulic retention time ranges from 45 to 60 days) [19], and depth-varying effluent quality (inadequate mixing) [20]. Furthermore, odor issues (e.g., sulfur dioxide) and the release of methane and other greenhouse gases affect the depletion of the ozone layer [21,22]. Due to the many

negative concerns associated with open ponding systems, researchers began to explore non-conventional POME treatment methods.

Non-conventional POME treatments are generally categorized into biological, physical, chemical, physicochemical, and bioelectrochemical (e.g., microbial fuel cell) processes [18]. The biological treatment methods, mainly aerobic and anaerobic biodegradation—especially anaerobic digestion—are efficient methods and have been widely explored (Table 2). Confined bioreactors such as advanced anaerobic expanded granular sludge bed (AnaEG) reactor, up-flow anaerobic sludge blanket-hollow centered packed bed reactor (UASB-HCPB) were engineered to treat POME while capturing biogas [23]. However, aerobic treatments such as rotating biological contractor (RBC), aerobic attached-growth system, aerobic oxidation, sequencing batch reactor (SBR), and moving bed biofilm reactor (MBBR) are rarely used to treat raw POME that has a high level of organic load [4].

**Table 2.** Performance of various anaerobic digestion closed systems on POME treatment [24].

| Method | COD Removal Efficiency |
| --- | --- |
| Anaerobic filtration | 94% |
| Up-flow anaerobic sludge blanket (UASB) | 98.4% |
| Anaerobic sequencing batch reactor (ASBR) | 92% |
| Expanded granular sludge bed (EGSB) | 94.89% |
| Continuous stirred tank reactor (CSTR) | 80% |
| Ultrasonic membrane anaerobic system (UMAS) | 93–98.7% |

Apart from biological treatments, physical and chemical treatment techniques are alternative means to treat POME as a polishing or post-treatment technique to meet the discharge water quality standards set by authorities. Adsorption technology using different types of media (e.g., activated carbon, fly ash, and zeolite) is used in POME treatment to remove residual oil and suspended and dissolved matter [10]. However, adsorption treatment needs additional pretreatment [25]. In the case of membrane separation/filtration treatment, high removal of solids can be obtained with good quality water output [26]. However, those membranes are prone to fouling, and these treatment methods require high capital and maintenance costs [27,28]. The advanced oxidation process is another post-treatment method that is very effective in treating POME, which can be used without producing sludge [29]. However, this is an energy and cost-intensive method [4]. Electrooxidation (EO) is categorized under advanced oxidation and has recently attached excessive attention owing to its eco-friendly treatment, high efficiency, and compatibility in implementation [30]. In EO, reactive oxygen species are produced that can mineralize dissolved organic compounds completely [31]. Nevertheless, EO is inefficient in treating suspended matters [32]. The coagulation and flocculation processes are other cheaper ways to treat POME; however, this process only reduces suspended solids in POME, and a large number of coagulants are required [33]. Electrocoagulation (EC)/electrocoagulation–flotation (ECF) with sacrificial anodes is another method that can be used as a post-treatment of biologically treated POME. It is a straightforward process, and high removal efficiency can be achieved within a minimum time [34,35]. However, high operating costs due to electrical energy consumption, regular replacement of sacrificial anode, and electrode passivation are some of the major limitations of EC [36].

Since the individual application of chemical coagulation and ECF to treat POME has been associated with many limitations, combined chemical coagulation and electroflotation (EF) with dimensionally stable electrodes may be a potential option to achieve maximum efficiency using their synergetic effect to sustainably treat POME while reducing specific energy consumption. The production of hydrogen gas ($H_2$) and oxygen gas ($O_2$) in EF promotes the flotation of flocs to separate contaminants from the water. Equations (1) and (2) show redox reactions, which occurred at the cathode and anode [37]. The reduction of water reaction takes place on the cathode resulting in the generation of $H_2$ and hydroxyl ions

(OH$^-$) (Equation (1)), and oxidation of water reaction takes place on the anode resulting in the generation of O$_2$ and hydrogen ions (H$^+$) (Equation (2)) [38,39].

At the cathode:

$$2H_2O_{(l)} + 2e^- \rightarrow H_{2(g)} + 2OH^-_{(aq)} \tag{1}$$

At the anode:

$$2H_2O_{(l)} \rightarrow O_{2(g)} + 4H^+_{(aq)} + 4e^- \tag{2}$$

The EF technique has many advantages over other conventional flotation methods: highly useful and a competitive alternative to settling tanks which require large land area [40], uniform mixing can be achieved owing to the production of gas bubbles [41], and EF units have a small compact that requires lower maintenance and running costs compared to other flotation units [40]. However, in the EF process, no sacrificial anode produces metallic ion coagulants in EF; therefore, coagulants should be added externally. Aluminum (Al)-based and iron (Fe)-based coagulants are generally used in wastewater and water treatment [42–44]. Polyaluminum chloride (PAC) is considered highly efficient than other coagulants owing to the presence of high-charged polymeric forms of Al hydrolysis products [43,45]. PAC is prepared by partial hydrolysis of acid aluminum chloride (AlCl$_3$) and generally consists of Al monomers, dimers, and trimers [43]. Cationic species absorb the negatively charged particles found in water and neutralize the charge based on the principle of charge neutralization [46]. In this mechanism, particles are destabilized, and subsequently, aggregation occurs [47]. In addition, coagulant aids/flocculants are often essential to further improve the performance of the coagulation process and reduce the residual Al in treated water [48]. Cationic polyacrylamides (CPAM) are a commonly applied flocculant in wastewater treatment [49,50]. As a result of high charge density, large molecular weight, and consisting of many functional groups, CPAM shows both charge neutralization and adsorption bridging functions simultaneously [50], which may help to generate stable and large quick-settling flocs during flocculation.

Since there is no or very little information found in the combined chemical and EF method as a post-treatment method for POME treatment, in this study, we hypothesized that combined chemical coagulation and EF treatment could significantly improve the treatment efficiency by reducing the specific energy consumption by limiting the electrolysis time. We used dimensionally stable electrodes such as platinized titanium anode and a stainless steel cathode for the EF.

## 2. Materials and Methods

### 2.1. Preparation of Synthetic POME

According to previous studies [51–54], concentrations of COD, total suspended solids, and O&G of POME have been reported in the range between 15,000–100,000 mg/L, 5000–54,000 mg/L, and 130–18,000 mg/L, respectively. The pH of POME normally varies between 4–5 [55]. In this study, a post-treatment method of POME was demonstrated. Previous studies showed that pretreated POME using biological treatment methods resulted in 4568 mg/L [56], 2431–2742 mg/L [57], and 1372 mg/L [58] COD concentrations. Therefore, we planned to prepare synthetic POME having a COD concentration in a range of 2500–4500 mg/L. Synthetic POME for the experiments was synthetically prepared using a soup palm base which was purchased from Nkulenu Industries Ltd. in Madina, Ghana. This is a pure and natural soup palm base, free from preservatives and coloring, which contains 77% palm fruit pulp, water, and salt. The POME was prepared by adding water to the palm fruit pulp (mesocarp) and boiling it for 30 min. Clean palm oil was extracted from the surface, while the sludge retained at the bottom was used as POME. This synthetically prepared POME was thick brownish–yellow in color and consisted of elevated levels of solid residues and unrecovered oil. The manually made POME was immediately transported to the Applied Ecological Engineering laboratory at Saitama University using polyethylene tanks. POME was diluted 10 times before every experiment to reduce the initial concentration to demonstrate the post-treatment influent of POME. The characteristics

of POME used in this study are given in Table 3, and the visual impression can be seen in Figure 1.

**Table 3.** Characteristics of synthetic POME prepared for the experiments.

| Parameter | Value (Average, SD) |
|---|---|
| pH | $6.64 \pm 0.1$ |
| Electrical conductivity | $451 \pm 20$ μS/cm |
| Oxidation–reduction potential (ORP) | $164 \pm 10$ mV |
| Turbidity | $1431 \pm 10$ NTU |
| Dissolved oxygen | $8.24 \pm 0.2$ mg/L |
| Dissolved hydrogen | <0.001 mg/L |
| COD | $3860 \pm 30$ mg/L |

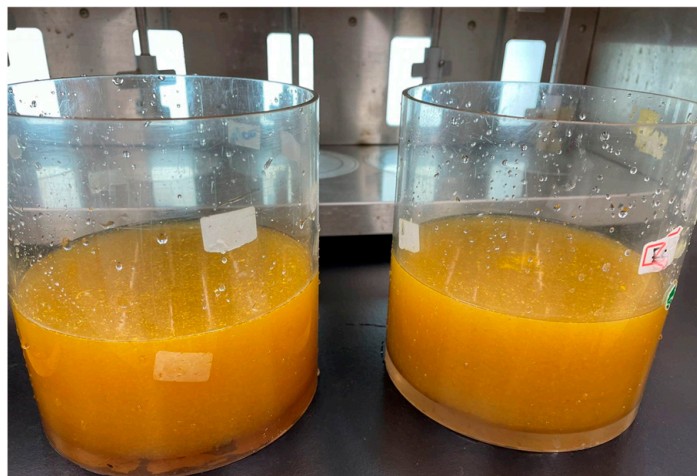

**Figure 1.** Appearance of synthetic palm oil prepared for the experiment.

### 2.2. Experimental Procedure

Experiments were carried out in three stages: EF treatments, chemical coagulation, and combined EF and chemical coagulation. Each set of experiments was carried out in triplicates at room temperature.

### 2.2.1. Electroflotation (EF)

As shown in Figure 2, a laboratory scale EF setup was made up of plexiglass, circular-shaped electrodes, and other accessories. A cylindrical platinized titanium anode and a cylindrical stainless steel cathode were connected to an external DC power supply (PMC35-2A, Kikusui, Kanagawa, Japan). Table 4 shows the characteristics of the EF setup used in this study.

**Table 4.** Characteristics of EF setup.

| | |
|---|---|
| Electrode material | Anode: Platinized titanium, Cathode: Stainless steel |
| Shape | Cylindrical |
| Effective surface area | Anode: 340 cm$^2$, Cathode: 396 cm$^2$ |
| Inter electrode distance | 1 cm |
| Reactor dimensions | 15 cm (Height); 16 cm (Diameter) |
| Effective volume of the cell | 1 L |

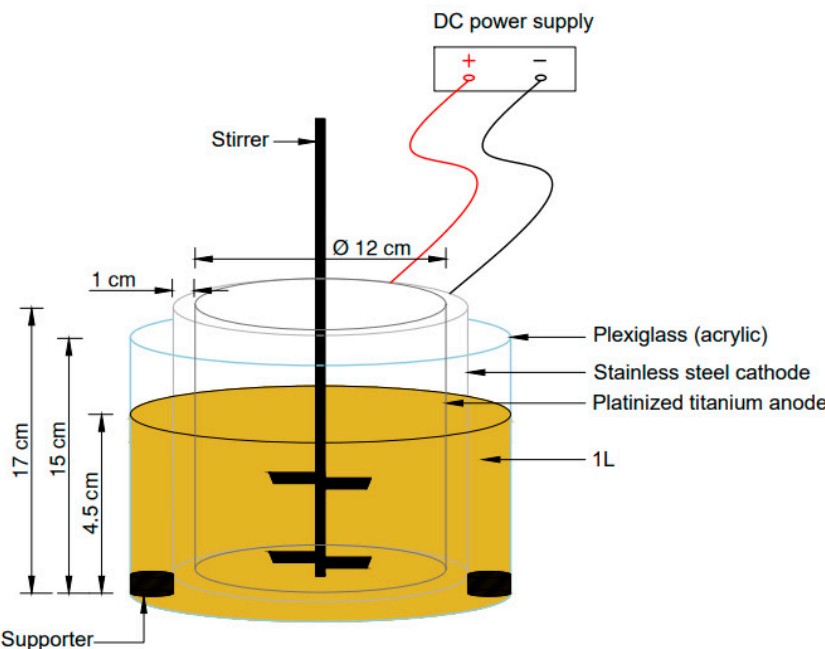

**Figure 2.** Schematic of EF setup for POME treatment. Equipped with a cylindrical-shaped platinized titanium anode, stainless steel cathode, and a DC power supply.

At the beginning of each EF test, 1 L of POME was fed into the EF cell, respectively. The effect of electrolysis operation time (0 to 45 min) and current intensity (1 A and 2 A) on POME treatment was monitored. Current densities at 1 A and 2 A were 2.5 and 5 mA/cm$^2$, respectively. A continuous stirring was applied at 120 rpm, and each set of experiments was performed at room temperature. Samples were collected at 0, 10, 15, 30, and 45 min. After settling for 10 min, turbidity, COD, zeta potential, DO (dissolved oxygen), and DH (dissolved hydrogen) of the liquid fraction were measured.

### 2.2.2. Chemical Coagulation

PAC and CPAM were used to demonstrate the effect of chemical coagulation on POME treatment. PAC, which contains 11% of Al$_2$O$_3$, was purchased from Nitto Chemical Industries, Ltd., Kanagawa, Japan. Commercially available CPAM was supplied by MT Aqua Polymer, Inc., Tokyo, Japan. The CPAM used in this study was prepared from the radical polymerization of acrylamide monomers. Characteristics of the CPAM used in this study were as follows: polyacrylamide polymer type was C-512; the principal component was polyacrylic acid ester type; the ionic characteristic was medium/high cationic, molecular weight was 4 million, and the viscosity was 230 mPas at 25 °C. A 2000 mg/L stock solution was prepared by dissolving 0.4 g of the polymer into 200 mL of distilled water at a temperature between 30–50 °C. The CPAM stock solutions were agitated at 300 rpm until the polymer particles were completely dissolved.

Laboratory scale chemical coagulation experiments were performed using jar test apparatus (JMD6E, Miyamoto Riken Ind. Co., Ltd., Osaka, Japan) using 1 L POME samples. For the coagulation and flocculation test, different dosages of PAC and CPAM (10, 20, 40, 60, and 100 mg/L) were used. After adding PAC, POME samples were rapidly stirred for 2 min at 120 rpm, followed by slow stirring for 3 min at 20 rpm after adding CPAM. At the end of the coagulation-flocculation process, the floc settled for about 10 min. After settling, turbidity, COD, and zeta potential were determined from the supernatant.

### 2.2.3. Combined Experiments

After determining the optimum electrolysis time and current intensity of EF, chemical coagulation and EF were combined. In this test, 15 min electrolysis time, at 1 A, was first carried out at 120 rpm, followed by adding different dosages of CPAM and PAC (20, 40, 60,

80, and 100 mg/L), respectively. After 10 min settling, turbidity, COD, and zeta potential were determined from the supernatant.

### 2.3. Analytical Methods

Treated water samples were collected after 10 min settling from 5 cm below the surface using a pipette, and the physicochemical parameters were analyzed as listed in Table 5.

**Table 5.** Physicochemical parameters and method adopted.

| Parameter (Units) | Instrument | Model |
|---|---|---|
| pH | Portable digital meter | HM-40P, DKK-TOA |
| Electrical conductivity (µS/cm) | Portable conductivity meter | AS710 |
| Zeta potential (mV) | Zeta potential and particle size analyzer | ELSZ-2000 |
| DO (mg/L) | Portable DO meter | HQ30D |
| DH (mg/L) | Portable hydrogen meter | ENH-2000 |
| Turbidity (NTU) | Laboratory turbidity meter | 2100 N |

COD was measured using the reactor digestion method with 50–500 mg/L range $COD_{Cr}$ vials. A 2 mL of homogenized sample was added into individual $COD_{Cr}$ test vials and incubated in a COD reactor (45600, HACH, Loveland, CO, USA) for 2 h at 150 °C. The COD readings were obtained using a photometer (Spectroquant NOVA 60, Merck, Darmstadt, Germany) after cooling to room temperature.

### 2.3.1. Removal Efficiency (R%)

The removal efficiencies (*R*%) have been calculated with Equation (3):

$$R\% = \frac{C_0 - C_1}{C_0} \times 100 \tag{3}$$

where $C_0$ and $C_1$ are concentrations of turbidity and COD before and after treatment, respectively.

### 2.3.2. Specific Energy Consumption

The specific energy consumption was calculated using kWh per unit mass of pollutant treated using Equation (4) [59,60].

$$\text{Specific energy consumption (kWh/g)} = \frac{V \times I \times t}{(C_0 - C_t) \times \forall} \tag{4}$$

where *V* is the average cell voltage (V), *I* is the applied current (amp), *t* is the electrolysis time (h), $\forall$ is the volume of the POME in electrolysis cell units (L), $C_0$ is the initial concentration of the pollutant (mg/L), and $C_t$ is the concentration at time t (mg/L).

### 2.3.3. Operational Cost

Energy and chemical costs were considered for the calculation of operating costs. Other costs were assumed to be fixed and were not included in the calculation (e.g., labor, electrode, and maintenance expenses). The operating costs for the treatment of POME were calculated according to Equation (5) [60].

$$\text{Operational cost} = aC_{energy} + bC_{coagulants} \tag{5}$$

where *a* is the unit cost of electricity (YEN/kWh) and b is the cost of coagulants (YEN/kg), and $C_{energy}$ and $C_{coagulant}$ are experimental, which are calculated per gram of COD treated.

### 2.4. Statistical Analysis

All graphs were produced using OriginLab software 2022 (OriginLab Corporation, Northampton, MA, USA). Statistical analyses were carried out using IBM SPSS Statistics 20.0 software. Turbidity and COD removal efficiencies under different treatments were compared using one-way ANOVA (analysis of variance) post-hoc Tukey HSD (Honestly significant difference) tests to identify the best treatment. All statistical analyses used a significance level of 5% ($p \leq 0.05$).

## 3. Results and Discussion

### 3.1. Effect of Electroflotation on POME Treatment

In the initial stage, the effect of electrolysis time and current intensity for POME treatment using EF were investigated using turbidity, COD, and zeta potential. The average initial turbidity and COD were 1431 ± 10 NTU and 3860 ± 30 mg/L, respectively. POME samples were treated using EF separately under 1 A and 2 A current intensities (current densities were 2.5 and 5 mA/cm$^2$, respectively). Figure 3 shows the variation of turbidity removal efficiency with the electrolysis time.

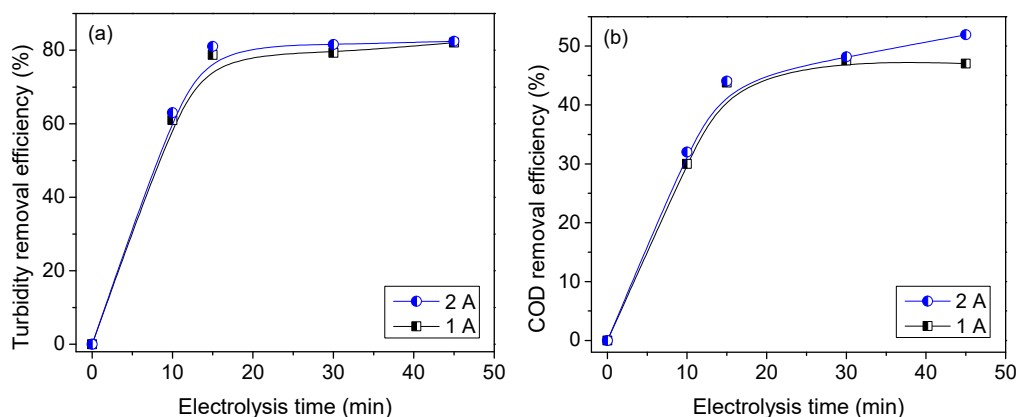

**Figure 3.** Comparison of (**a**) turbidity and (**b**) COD removal efficiency between 1 A and 2 A current intensities under EF for POME treatment.

From the results, it could be realized that increasing electrolysis time and current density increased the turbidity and COD removal efficiency. This is because bubble size, generation rate, and collisions among particles depend on Faraday's law. High electrical current resulted in significant bubble density smaller in size, which promotes the removal of pollutant in EF treatment [38] through the attachment of suspended pollutants to the charged surface of micro-bubbles [61].

In EF treatment, H$_2$ (Equation (1)) and O$_2$ (Equation (2)) are produced at the cathode and anode, respectively. Results showed that DH was increased from <0.001 to 0.24 mg/L after 45 min of electrolysis time at 2 A as a result of increasing electrolysis time and current density. This was also reported by Trinke et al. [62]. They also noticed that the higher the DH evolution because of increasing applied current density, thus leading to a rising supersaturation of DH. In this study, it was noticed that DO concentration also increases at 2.5 mA/cm$^2$ current density where the current intensity was maintained at 1 A; however, it later decreased when the current density was increased to 5 mA/cm$^2$ at 2 A. The research carried out by Ben et al. [63] also supports this, as it was reported that at higher current densities, a larger amount of O$_2$ was inducted into the liquid phase, resulting in more gas bubbles. This increased DO concentration caused an increase in oxygen permeation. Thus, in this study, it could be said that increasing electrical current brought about greater bubble generation, which effectively promoted pollutant removal by EF.

Even though the COD removal efficiency continued to increase at 2A, there was no significant variation of turbidity removal efficiency after 15 min either at 1 or 2 A. The turbidity and COD removal efficiency after 15 min electrolysis were 79% and 44%,

respectively, which was achieved at 1 A current intensity. Similar results were reported by Ho and Chan [64]. They demonstrated the EF treatment with a lead dioxide–titanium anode and a stainless steel cathode for POME treatment and achieved about 86% of suspended solids and 40% of COD removal efficiency.

In this study, we observed a gradual increase in COD removal efficiency at 2 A with increasing the electrolysis time. In other words, when the current intensity and electrolysis time increased from 1 to 2 A and 15 to 45 min, respectively, the COD removal efficiency increased from 44% to 52%, whereas specific energy consumption increased from 0.0052 kWh to 0.026 kWh for the treatment of one gram of COD.

Increasing the COD removal efficiency could be a result of the production of reactive oxygen species at platinized titanium anode, such as hydroxyl radicals (HO°), which mineralize the organic pollutants [59]. This phenomenon is known as EO. In the EO process, HO° are produced on the anode surface at a high current (Equation (6)).

$$M + H_2O_{(l)} \rightarrow M + \left(HO^\circ\right) + H^+_{(aq)} + e^- \tag{6}$$

Zeta potential is a crucial and controlling parameter in wastewater treatment studies, especially in coagulation and flocculation, because it measures the magnitude of electrostatic charges of particles in the system. As shown in Figure 4, the average initial zeta potential was $-14.62 \pm 0.8$ mV. The zeta potential was gradually increased with electrolysis time in both current intensities owing to the destabilization of suspension caused by particle aggregation as a result of the generation of gas bubbles [47]. However, even after 45 min electrolysis time zeta potential of POME samples treated under 1 A and 2 A current were $-12.76 \pm 0.7$ and $-9.97 \pm 0.4$ mV, respectively. This is because gas bubbles generated between pH 7–8 may have negative zeta potential [61]. Therefore, the application of coagulants could be an appropriate option to improve the zeta potential toward its positive shift.

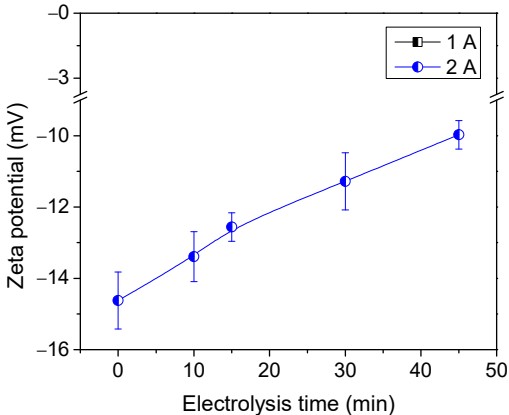

**Figure 4.** Variation of zeta potential with treatment time at different current intensities in EF treatment. Error bars represent the standard deviation (*n* = 3).

### 3.2. Effect of Chemical Coagulation on POME Treatment

In this section, the effect of PAC and CPAM dosage on POME treatment was investigated using turbidity and COD removal efficiency and variation of zeta potential. Figure 5 shows the turbidity and COD removal efficiency under different chemical treatments. The removal efficiency increased when the chemical dosage was increased. Continual addition of PAC dosage until 100 mg/L showed 25% of COD and 94% of turbidity removal efficiency. The addition of CPAM dosage until 100 mg/L also enhanced turbidity and COD removal efficiency up to 96% and 34%, respectively. The optimum turbidity and COD removal efficiencies were 90% and 47%, reported at combined PAC and CPAM treatment at a dosage of 20 mg/L PAC and CPAM.

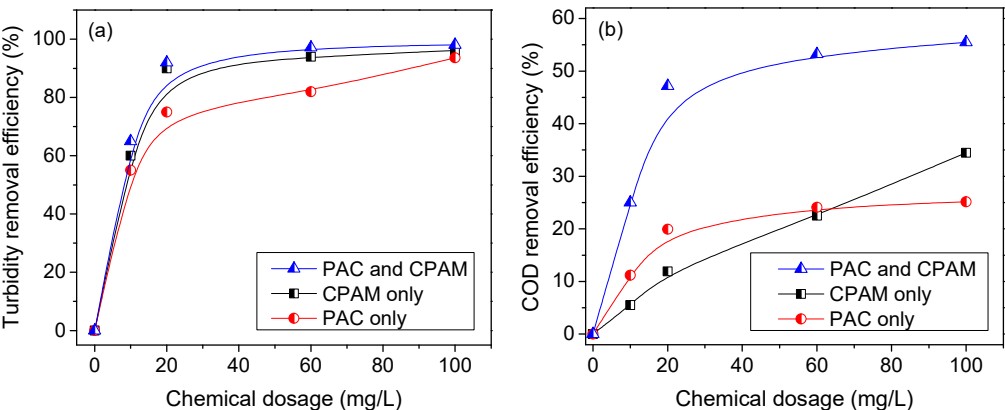

**Figure 5.** Comparison of (**a**) turbidity and (**b**) COD removal efficiency at different chemical coagulation methods for POME treatment.

When 40 mg/L of PAC dosage was used with a varying dosage of CPAM, organic pollutant removal was higher. The maximum removal efficiency was observed to be 55% and 98% for COD and turbidity, respectively, at 100 mg/L CPAM dosage.

Figure 6 shows the variation of zeta potential with chemical dosage at different treatments. At PAC and combined PAC and CPAM treatments, the zeta potential was proportionally increased when the chemical dosage was increased. The final average zeta potential of PAC and CPAM treatments were $-5.97 \pm 0.5$ and $-8.65 \pm 0.6$ mV, respectively. In the CPAM treatment, the zeta potential was slightly increased until 60 mg/L dosage, and it was no longer increased after 60 mg/L.

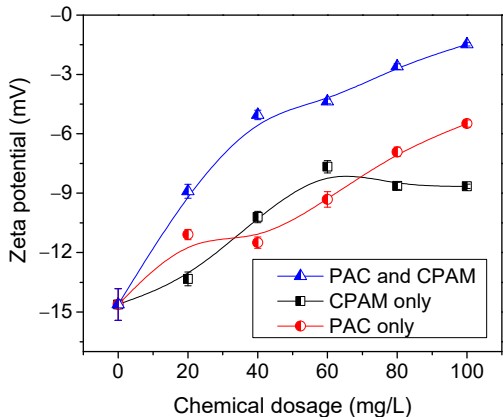

**Figure 6.** Variation of zeta potential with the treatment time at different chemical coagulation methods. Error bars represent the standard deviation ($n = 3$).

In PAC treatment, highly stable, +7 charge $AlO_4 Al_{12}(OH)_{24}^{7+}$ species are pre-formed and rapidly neutralize negatively charged suspended pollutants in POME, subsequently, zeta potential increased [65]. It has also been observed that increasing cationic coagulant dosage increased charge neutralization in the treated water. The final average zeta potential of combined PAC and CPAM treatments was $-2.76 \pm 0.4$ mV. This is because PAC and CPAM both have cationic properties, and their combined effect on pollutant removal is enhanced in accordance with the principle of charge neutralization. Increasing CPAM dosage increased the intrinsic viscosity, which led to molecular chain growth. In addition, CPAM consists of a special linear structure and abundant functional groups, which help to attract various pollutants [66]. Furthermore, increasing the CPAM dosage also increased the probability of collision among the colloidal particles and favored the trapping and bridging of particles [67]. These changes can be attributed to the increased cationic effect

of PAC and CPAM that caused an increase in aggregation and enhanced colloidal organic matter removal in POME treatment (Figure 7).

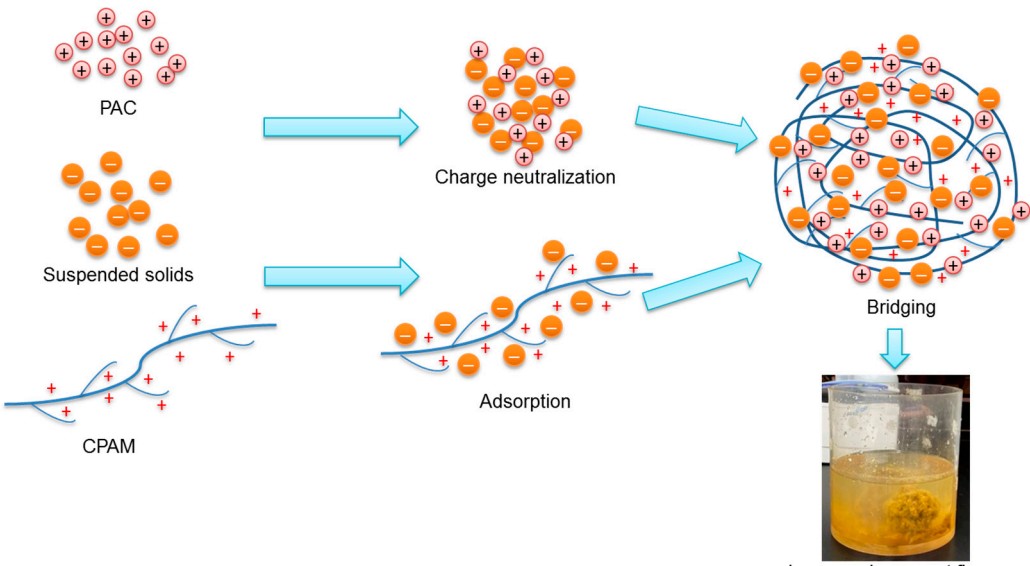

**Figure 7.** Mechanisms of coagulation and flocculation.

### 3.3. Effect of Combined Chemical Coagulation and Electroflotation on POME Treatment

In this section, experiments were carried out using a fixed electrolysis time of 15 min and a current density of 2.5 A/cm$^2$ at 1 A current in the EF system, a fixed dosage of PAC 40 mg/L, and varying dosages of CPAM.

When chemical coagulation (PAC and CPAM) was combined with EF, COD removal efficiency increased from 55% to 60% (Figure 8). Overall, the highest turbidity and COD removal efficiency of 97% and 60% were noticed when 40 mg/L of PAC and 100 mg/L of CPAM were used with EF. However, no significant removal efficiency was observed in terms of turbidity or COD after 20 mg/L of CPAM. Turbidity and COD removal efficiency at 20 mg/L of CPAM dosage were 96% and 54%, respectively. Therefore, 20 mg/L was selected as the optimum concentration for CPAM considering the cost and residual impact. Releasing residual CPAM into the environment might be toxic for aquatic life [68].

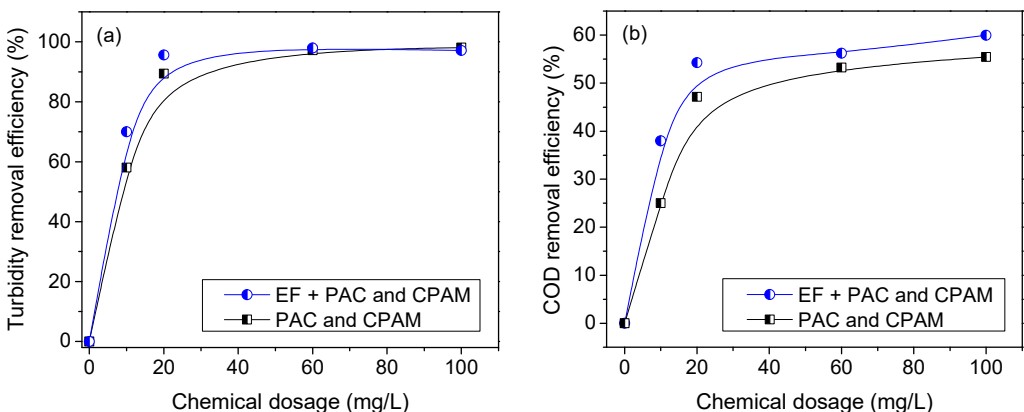

**Figure 8.** Comparison of (**a**) turbidity and (**b**) COD removal efficiency between EF and combined EF and chemical coagulation for POME treatment.

As shown in Figure 9, both treatments showed an increase in zeta potential when the chemical dosage was increased. There was no significant impact of EF on increasing the

zeta potential. Combined chemical and EF treatment improved the zeta potential up to −2.97 ± 0.2 mV; however, it was −12.76 ± 0.7 mV when POME was treated with EF at 1 A.

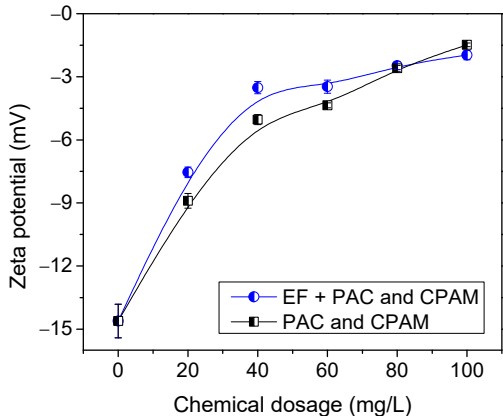

**Figure 9.** Variation of zeta potential with the treatment time at chemical coagulation and combined EF and chemical coagulation. Error bars represent the standard deviation (*n* = 3).

Figure 10 shows the appearance of synthetic POME after different treatment combinations. One-way ANOVA results revealed that there was a statistically significant difference in turbidity ($F_{(2,6)}$ = 51.485, *p* = <0.001) and COD ($F_{(2,6)}$ = 17.007, *p* = 0.003) removal efficiency among different POME treatments (Figure 11). Tukey's HSD test for multiple comparisons found that the removal efficiency at combined chemical and EF treatment was significantly higher for turbidity (*p* < 0.001) and COD (*p* = 0.003) compared with EF treatments alone. Similarly, treatment efficiency at combined chemical and EF treatment was significantly higher for turbidity (*p* = 0.012) and COD (*p* = 0.018) compared with chemical coagulation treatments alone. Since combined experiments were performed at 1 A for 15 min electrolysis, specific energy consumption was reduced to 0.004 kWh for the treatment of one gram of COD.

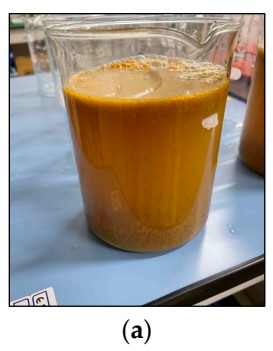
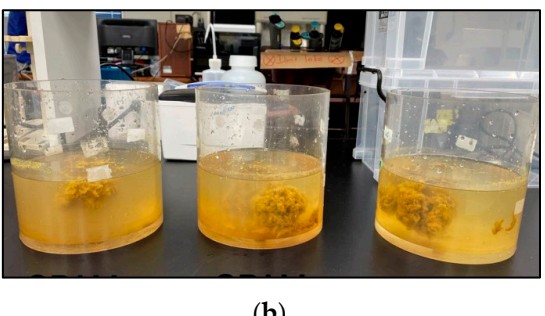
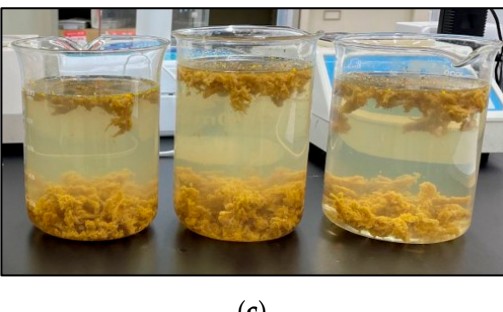

(**a**)　　　　　　　　　　　(**b**)　　　　　　　　　　　(**c**)

**Figure 10.** Appearance of synthetic POME after treatment with (**a**) EF, (**b**) PAC + CPAM, and (**c**) EF + PAC + CPAM.

These results proved the impact of applying PAC and CPAM on EF treatment to improve the efficiency and reduce the specific energy consumption in POME treatment. Furthermore, the results of operational cost calculation proved that the combined EF and chemical treatment is economically feasible for the post-treatment of POME. When EF was combined with chemical treatment, the electrical energy cost for treating one kilogram of COD was reduced from YEN 378 to YEN 108 as a result of reducing the current density from 5 to 2.5 mA/cm² and treatment time by 30 min (45 min to 15 min). However, an additional cost of YEN 22 was included for PAC and CPAM. Therefore, the total cost for treating one kilogram of COD when EF was combined with chemical treatment can be calculated as YEN 130.

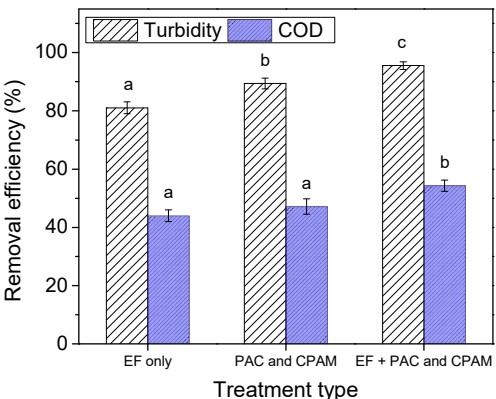

**Figure 11.** Comparison of different treatment types. Error bars represent the standard deviation (*n* = 3). Different lower-case letters above the bars indicate the significant difference among treatment types separately in turbidity and COD removal efficiency at a significance level of 0.05.

## 4. Conclusions

In this study, we demonstrated that the combined chemical and EF treatment with a dimensionally stable platinized titanium anode and a stainless steel cathode could be a viable and efficient option for the post-treatment of POME. The production of $H_2$ and $O_2$ at EF promotes the separation of flocs produced by chemical coagulation. This method resulted in 96% and 54% of turbidity and COD reduction, respectively, within 15 min electrolysis at 1 A current intensity when EF was combined with PAC at a dosage of 40 mg/L together with CPAM at a dosage of 20 mg/L. The specific energy consumption was reduced by 71% when EF was combined with chemical coagulation as a result of the reduction in the current and electrolysis time. However, COD in the treated water did not meet the discharge water quality standards. Therefore, further studies should be focused on EO for the mineralization of dissolved organic matter to meet the discharge water quality standards given by the authorities. Furthermore, certain limitations including requirement of longer treatment time for EF at low current densities, difficulty in maintaining an appropriate conductivity, and passivation of electrodes of this method may affect scale-up and operational activities. However, scaling up these laboratory-scale setups to pilot-scale models upon further research will be extremely useful for subsequent industrial-scale implementations.

**Author Contributions:** Conceptualization, E.Y.F. and T.F.; methodology, E.Y.F. and T.F.; investigation, E.Y.F.; data analysis, E.Y.F. and T.A.O.K.M.; writing—original draft preparation, E.Y.F., T.A.O.K.M. and T.F.; writing—review and editing, T.A.O.K.M. and T.F.; visualization, T.F.; supervision, T.F. All authors have read and agreed to the published version of the manuscript.

**Funding:** This study was supported by the Japan Society for the Promotion of Science (JSPS) KAK-ENHI (22K12472) and a research grant from the Strategic Research Area for Sustainable Development in East Asia (SRASDEA), Saitama University.

**Institutional Review Board Statement:** Not applicable.

**Informed Consent Statement:** Not applicable.

**Data Availability Statement:** Not applicable.

**Acknowledgments:** The authors would like to acknowledge the graduate assistance scheme of Saitama University for the financial support, Ken Takeda, MT Aqua Polymer, Inc., Japan, for supplying cationic polyacrylamide polymer, Toshinori Takahashi and Yi Zhang for their support in the laboratory.

**Conflicts of Interest:** The authors declare no conflict of interest.

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
