# Peer review of "Synergetic Effect of Chemical Coagulation and Electroflotation on Synthetic Palm Oil Mill Effluent Treatment"

_2673-4079, doi:10.3390/suschem4020010_

Round 1
Reviewer 1 Report
In this study, authors discuss Synergetic Effect of Chemical Coagulation and Electroflotation on Pseudo Palm Oil Mill Effluent Treatment. Paper has good quality to publication after below correction:
- Abstract isnot acceptable and must be improve with more data
- Abbreviation must be remove from keyword
- Introduction must be start with a sentence about environmental pollutants and why removal of them are important and confirm with Environmental Research Volume 222, 1 April 2023, 115338 and Environmental research 207(2022) 112156
- purity materials must be clear
- number of experimental relative to error bar must be add
Author Response
Reviewer 1
Response table
|
No. |
Review comment |
Response |
|
1 |
Abstract is not acceptable and must be improve with more data |
Thank you very much for your valuable comments. These comments were really helpful to improve our manuscript. We added more results to the abstract and it was modified (Lines 24–26, 32). |
|
2 |
Abbreviation must be remove from keyword |
"POME" abbreviation was removed (Line 36). |
|
3 |
Introduction must be start with a sentence about environmental pollutants and why removal of them are important and confirm with Environmental Research |
The following sentence was added. "Indiscriminate disposal of high-strength organic pollutants poses a threat to the surrounding water bodies, leading to eutrophication and endangering aquatic life. Palm oil mill effluent (POME) is, which contains large quantities of organic matter, considered the most environmentally harmful when discharged untreated (Lines 40–43). |
|
4 |
Purity materials must be clear |
We are not sure about the meaning of the question exactly. However, we mentioned the purity and other details of the materials used in this study (e.g. PAC, CPAM) (Lines 254–264). |
|
5 |
Number of experimental relative to error bar must be add |
Each set of experiments was carried out in triplicates and the average values were used to calculate the removal efficiency of turbidity, COD, etc. Therefore, efficiency graphs do not show SD values. |
Reviewer 2 Report
The manuscript provides a nice discussion on synergetic effect of chemical coagulation and electroflotation (EF) on pseudo–Palm Oil Mill Effluent (POME) treatment. The topic is worthy of scientific investigation and the manuscript would be suitable for consideration for publication in the field of sustainability. Furthermore, it has significant industrial relevance, in view of the sizeable global palm oil market and the environmental impact due to POME. As the manuscript points out insightfully, every ton of crude palm oil can generate around 2.5 to 3.8 tons of POME as part of industrial processing. There are, however, a few points to address and that may require revision to further strengthen the manuscript.
1)
To enhance readability, please carry out proof reading of the whole manuscript to work on linguistic expressions and correct for inadvertent typo errors.
For example, Page 1 (Abstract) Line13: it should be “…harmful by-products and ease of operation”, and not “…harmful bi-products and ease in operation”.
2)
Page 2-3 (lines 65 – 102):
It is commendable that the manuscript provides a review of the various technologies used in the treatment of POME. To facilitate reading, the information provided can be categorised and presented in the form of a table by comparing the different treatment technologies (e.g. lagoon, anaerobic processes, RBC, SBR, MBBR, membrane separation, adsorption, advance oxidation, coagulation and flocculation, EF, ECF, etc..) against the evaluation criteria such as operating costs, specific energy consumption, specific chemical consumption, removal efficiency, footprint, technology readiness level, case studies of pilot or demonstration plants, etc..
3)
Page 5 (Table 2):
The lab-scale reactor is small at about 1 L – the manuscript should clarify what are the limitations of the study and precautions to take due to the relatively small scale of the reactor system to ensure accurate and reliable results.
4)
Page 6 (Equation 5):
The units on both sides of the equation are not consistent. On the left-hand side of the equation, it is kWh/m3 (denominator is volume and in cubic metre). However, on the right-hand side of the equation, it is kWh/mg (denominator is mass and in milligram). Please check the equation carefully and revise subsequent analyses accordingly.
5)
Page 7 (Lines 335 – 339):
To help in understanding, please provide a figure or drawing to clearly illustrate the underlying scientific mechanisms as described in words:
“…Increasing CPAM dosage increased the intrinsic viscosity which led to molecular chain growth. In addition, CPAM consists with a special linear structure and abundant functional groups, which help to attract various pollutants [50]. Furthermore, increasing CPAM dosage also increased the probability of collision among the colloidal particles and favored trapping and bridging…”
6)
Page 12 5. Conclusions (Lines 395 – 397):
It is written that “…However, either treated water turbidity or COD did not meet the discharge water quality standards. Therefore, further studies should be focused on EO for the mineralization of dissolved organic matter to fulfil the discharge water quality standards given by authorities.”
Please revise and provide a more detailed discussion to cover the following points:
- - Please state what exactly are the discharge water quality standards given by which authorities?
- - If the treated water is not able to meet the discharge standards, what further technologies will be needed for post-treatment?
- - Please elaborate on what is the roadmap to further develop this technology to scale it up for full implementation?
Author Response
eviewer 2
Response table
|
No. |
Comment |
Response |
|
1 |
To enhance readability, please carry out proof reading of the whole manuscript to work on linguistic expressions and correct for inadvertent typo errors. For example, Page 1 (Abstract) Line13: it should be “…harmful by-products and ease of operation”, and not “…harmful bi-products and ease in operation”. |
Thank you very much for your valuable comments. These comments were really helpful to improve our manuscript.
The manuscript was checked for formatting errors and English. |
|
2 |
Page 2-3 (lines 65 – 102): It is commendable that the manuscript provides a review of the various technologies used in the treatment of POME. To facilitate reading, the information provided can be categorised and presented in the form of a table by comparing the different treatment technologies (e.g. lagoon, anaerobic processes, RBC, SBR, MBBR, membrane separation, adsorption, advance oxidation, coagulation and flocculation, EF, ECF, etc..) against the evaluation criteria such as operating costs, specific energy consumption, specific chemical consumption, removal efficiency, footprint, technology readiness level, case studies of pilot or demonstration plants, etc.. |
Anaerobic treatment methods are commonly used to treat POME and capture biogas. We compared the performance of various anaerobic digestion closed systems in terms of COD removal on POME treatment in Table 1. However, aerobic treatment is used as a tertiary treatment of POME. See Lines 119–120, 129–150. |
|
3 |
Page 5 (Table 2): The lab-scale reactor is small at about 1 L – the manuscript should clarify what are the limitations of the study and precautions to take due to the relatively small scale of the reactor system to ensure accurate and reliable results. |
To ensure the accuracy and reliability of results, triplicates were carried out and the reproducibility of results was ensured (233–234). Each set of experiments was carried out at room temperature (Line 235). pH and conductivity were maintained constant at the beginning of each experiment (Line 223, Table 2). The position of stirrer was positioned at a constant depth for each set of experiments to ensure the constant stirring mechanism. |
|
4 |
Page 6 (Equation 5): The units on both sides of the equation are not consistent. On the left-hand side of the equation, it is kWh/m3 (denominator is volume and in cubic metre). However, on the right-hand side of the equation, it is kWh/mg (denominator is mass and in milligram). Please check the equation carefully and revise subsequent analyses accordingly. |
The unit of the equation was changed. It was a mistake. It should be kWh/g. We changed accordingly (Line 297). |
|
5 |
Page 7 (Lines 335 – 339): To help in understanding, please provide a figure or drawing to clearly illustrate the underlying scientific mechanisms as described in words: “…Increasing CPAM dosage increased the intrinsic viscosity which led to molecular chain growth. In addition, CPAM consists with a special linear structure and abundant functional groups, which help to attract various pollutants [50]. Furthermore, increasing CPAM dosage also increased the probability of collision among the colloidal particles and favored trapping and bridging…” |
A schematic representation of the process was prepared (Lines 423–425, Figure 7). |
|
6 |
Page 12 5. Conclusions (Lines 395 – 397): It is written that “…However, either treated water turbidity or COD did not meet the discharge water quality standards. Therefore, further studies should be focused on EO for the mineralization of dissolved organic matter to fulfil the discharge water quality standards given by authorities.” Please revise and provide a more detailed discussion to cover the following points: · Please state what exactly are the discharge water quality standards given by which authorities? · If the treated water is not able to meet the discharge standards, what further technologies will be needed for post-treatment? · Please elaborate on what is the roadmap to further develop this technology to scale it up for full implementation? |
POME discharge quality standards were mainly established by the Department of Environment (DOE), Malaysia under the regulation of crude palm oil, in 1982. We also found some standards from Indonesia and Thailand. Those details were incorporated in the manuscript (Lines 81, Table 1 and Lines 83–87).
Since the proposed method is unable to fulfill the discharge water quality standards, we recommended further studies to combine electrooxidation treatment with the current method to enhance the removal of dissolved organic compounds presence in POME (Lines 495–496). There are limitations to be addressed in order to scale it up and implementation (Lines 497–499). Scaling up these laboratory-scale setups to pilot-scale models will be the next step, which will be truly useful for subsequent industrial-scale implementations (Lines 499–500). |
Reviewer 3 Report
To use of synthetic wastewater is always a bit challenging. At least its quality in the original wastewater should be verified based on the literature.

Author Response
Response table
|
No. |
Comment |
Response |
|
1 |
To use of synthetic wastewater is always a bit challenging. At least its quality in the original wastewater should be verified based on the literature. |
Thank you very much for your valuable comments. These comments were really helpful to improve our manuscript. The details of the original POME were added (Lines 202–209). |
|
General comments |
||
|
2 |
“Sustainable” as a word covers social, environmental and economic issues. This paper covers only environmental one. So, I strongly recommend that also economic issue will be take care of that we understand the “about” costs effects between examined alternatives. However, if this is against the policy of journal then ok for me. |
In this study, we showed the energy reduction of POME treatment by combining electroflotation with chemical treatment. This method will be economically feasible as we were able to reduce the energy consumption of electrofloatiaion by 71%. We added operational cost calculation results. Lines 303–313 showed the equation and its details. Lines 475–482 showed the respective results. |
|
3 |
PAC and CPAM were used in this research as additional chemicals. However, CPAM “Cationic Polyacrylamide Polymer” is not so sustainable (fossil based and poor biodegradability). This should be more explained why it was also used in this study. |
In this study, we used a very low dosage (20 mg/L) However, we further optimize the dosage of CPAM in our future studies and toxicity tests with plant-based bioassay will be carried out in order to measure the toxicity of residual CPAM and PAC. |
|
4 |
The use of this method do not meet guidelines set by authorities so it is pure scientific paper so far. |
Treated synthetic POME using this method did not meet discharge quality standards. However, further studies will be carried out chemical+Electroflotation coupled with electrooxidation to reduce dissolved organic compounds. |
|
5 |
Since I am not a native English speaker, I cannot comment on the quality of the language, but must do so on behalf of the magazine. However, I think it is very smooth to read and understand. |
We have again checked the manuscript for formatting errors and English. |
|
Detailed comments |
||
|
6 |
Line 11: “discharge water quality standards set by authorities” this will be given in the paper that we understand that how far behind we are those in this research. At least the literature source. |
POME discharge quality standards established by Malaysia, Indonesia, and Thailand were added to the manuscript (Lines 81–87). |
|
7 |
Lines 37, 38, 39 and 40: “The demand on palm oil is rapidly increased owing to its positive health impacts such as increasing brain health, decreasing the cholesterol level, reducing the oxidative stress, improving hair and skin health, etc. [2]”. On the other hand, the fat composition of palm oil is harmful to humans according to some studies. Should something be mentioned about this? |
The following sentence was added. "while some studies reported that the consumption of palm oil increases low-density lipoprotein cholesterol" (Lines 50–51). |
|
8 |
Lines 44, 45, 46 and 47 the characteristic of POME is listed, but same values should be mentioned here also than you mentioned in table 1, that we can estimate how your synthetic water differ from mill waters. |
In this study, we targeted to demonstrate a post-treatment method of POME. Accordingly, we prepared synthetic POME based on the actual POME values taken from the literature (Lines 202–209). |
|
9 |
Lines 81 and 82. See comment Line 11. |
See Lines 81–87. |
|
10 |
Line 161: “are given in Table 1.” like this: … are given in Table 1 and the visual impression can be seen in Photo 1. |
Modified according to the comment (Lines 221–222). |
|
11 |
Line 162: Temperature (T) should be also mentioned in Table 1 since it has effect on the viscosity of water at least. What might be the temperature of wastewater in real mill? If not measured should be said that made at room temperature. |
The temperature was not measured. We can understand that temperature is an important parameter for electrocoagulation. We will incorporate those measurements in our future studies. |
|
12 |
Line 218: T measurement unit should be mentioned here if measured. |
The temperature was not measured. |
|
13 |
The entries for figures 6, 7 and 8 / photo 2 are missing from the text. Those should be added to text before them like done earlier |
Those figures were referred in the text (Lines 434, 443, 463, and 466). |
|
Hint for economic evaluation if Journal accept |
||
|
14 |
pls. use conservative values not exact. Main principle is, that we can see is it economically feasible to use this kind combination for wastewater purification or not. |
We added operational cost calculation results. Lines 303–313 showed the equation and its details. Lines 475–482 showed the respective results. |